# Unraveling MASLD: The Role of Gut Microbiota, Dietary Modulation, and AI-Driven Lifestyle Interventions

**DOI:** 10.3390/nu17091580

**Published:** 2025-05-04

**Authors:** Carolina Jiménez-González, Marta Alonso-Peña, Paula Argos Vélez, Javier Crespo, Paula Iruzubieta

**Affiliations:** 1Gastroenterology and Hepatology Department, Clinical and Translational Research in Digestive Diseases, Valdecilla Research Institute (IDIVAL), Marqués de Valdecilla University Hospital, 39011 Santander, Spain; carolina.jimenez@idival.org (C.J.-G.); marta.alonso@unican.es (M.A.-P.); paula.argos@idival.org (P.A.V.); p.iruzubieta@gmail.com (P.I.); 2Departamento de Anatomía y Biología Celular, Universidad de Cantabria, 39011 Santander, Spain

**Keywords:** MASLD, gut microbiota, multi-omics, dietary patterns, lifestyle modifications, artificial intelligence

## Abstract

Gut microbiota has a crucial role in the pathophysiology of metabolic-associated steatotic liver disease (MASLD), influencing various metabolic mechanisms and contributing to the development of the disease. Dietary interventions targeting gut microbiota have shown potential in modulating microbial composition and mitigating MASLD progression. In this context, the integration of multi-omics analysis and artificial intelligence (AI) in personalized nutrition offers new opportunities for tailoring dietary strategies based on individual microbiome profiles and metabolic responses. The use of chatbots and other AI-based health solutions offers a unique opportunity to democratize access to health interventions due to their low cost, accessibility, and scalability. Future research should focus on the clinical validation of AI-powered dietary strategies, integrating microbiome-based therapies and precision nutrition approaches. Establishing standardized protocols and ethical guidelines will be crucial for implementing AI in MASLD management, paving the way for a more personalized, data-driven approach to disease prevention and treatment.

## 1. Introduction

Metabolic dysfunction-associated liver disease (MASLD), characterized by intrahepatic fat accumulation, is the most prevalent chronic liver disease in Western countries, with a global estimated prevalence of 38% [1]. The MASLD spectrum includes steatosis, metabolic dysfunction-associated steatohepatitis (MASH), fibrosis, cirrhosis, and hepatocellular carcinoma (HCC) [2]. Although not all cases progress to cirrhosis, it is estimated that about 20% of MASLD cases develop MASH, with potential advancement to cirrhosis and HCC [3,4]. Beyond liver-specific complications, MASLD significantly contributes to cardiovascular disease, chronic kidney disease, and certain cancers—positioning it as a key mediator of systemic morbidity.

MASLD arises from a complex interplay of metabolic risk factors, genetic susceptibility, environmental exposures, lifestyle, and gut microbiota. The intricate interplay among these factors has led to the conceptualization of MASLD not merely as a disease but as a syndrome.

In 2024, Resmetirom received accelerated FDA approval for patients with MASLD and significant fibrosis, following promising—though modest—histological improvements observed in the interim analysis of a phase 3 clinical trial [5]. Despite this advancement, lifestyle modifications remain the cornerstone of MASLD management [6]. Weight loss achieved through dietary changes and increased physical activity has been consistently linked to reductions in hepatic steatosis, inflammation, fibrosis, and cardiometabolic risk factors [7]. Notably, a weight loss of ≥10% is associated with MASH resolution and fibrosis regression [8]. However, only a minority of patients manage to achieve and sustain this level of weight reduction, underscoring the urgent need for more effective and sustainable therapeutic strategies. In recent years, the gut–liver axis has emerged as a central player in MASLD pathophysiology [9,10,11,12]. Distinct alterations in gut microbiota (GM) composition have been associated with hepatic fat accumulation, inflammation, and fibrosis progression. Moreover, microbial metabolites such as short-chain fatty acids (SCFAs), bile acids (BAs), and trimethylamine N-oxide (TMAO) have been shown to influence lipid metabolism, insulin sensitivity, and hepatic inflammation [13]. High-throughput omics technologies are advancing our understanding of microbial metabolites and aiding in the identification of novel biomarkers for MASLD [14]. The composition and function of the GM are strongly shaped by dietary patterns. While Western-style diets promote dysbiosis and disease progression, fiber-rich and plant-based diets (e.g., Mediterranean diet) support a more favorable microbial environment that may protect against MASLD. These insights have stimulated interest in microbiome-targeted interventions—including prebiotics, probiotics, synbiotics, and fecal microbiota transplantation (FMT)—as emerging tools in MASLD therapy [15].

Artificial intelligence (AI) is revolutionizing healthcare by enhancing disease diagnosis, treatment, and prevention. A key application is promoting lifestyle modifications through AI-driven interventions that analyze large-scale data, provide personalized recommendations, and optimize dietary guidelines [16,17]. Machine learning (ML) models have shown promise in predicting conditions like obesity, hypertension, type 2 diabetes mellitus (T2DM), and cardiovascular diseases. AI is also being explored in MASLD to improve patient stratification and biomarker discovery through automated data analysis [18]. The integration of microbiome data and blood metabolites could enhance personalized treatment strategies. Future work should focus on standardizing AI tools and validating their impact on long-term liver outcomes, gut microbiota modulation, and precision nutrition strategies. This review explores the complex interplay between gut microbiota and MASLD, with a particular focus on how dietary modulation can reshape microbial communities and influence disease progression. Additionally, we examine the emerging role of AI-driven tools in supporting lifestyle modifications, enabling precision nutrition, and optimizing MASLD management.

## 2. Gut Dysbiosis and Its Association with MASLD

### 2.1. Human Gut Microbiota

The human gut microbiome is a complex and dynamic ecosystem composed of bacteria, archaea, viruses, and eukaryotic microbes that inhabit the gastrointestinal tract. It plays a crucial role in maintaining physiological homeostasis and establishing a symbiotic relationship with the host [19]. The composition and relative abundance of microbial populations fluctuate throughout an individual’s life, influenced by genetic and environmental factors such as diet, medication, and microbial interactions [20].

Recent taxonomic revisions have led to updates in the phylum nomenclature to better reflect phylogenetic relationships, as recommended by the List of Prokaryotic Names with Standing in Nomenclature (LPSN) [21]. The predominant bacterial phyla in a healthy adult gut are *Bacillota* (formerly known as *Firmicutes*) and *Bacteroidota* (formerly *Bacteroidetes*), followed by *Actinomycetota* (formerly *Actinobacteria*), *Pseudomonadota* (formerly *Proteobacteria*), *Fusobacteriota* (formerly *Fusobacteria*), and *Verrucomicrobiota* (formerly *Verrucomicrobia*), which are present in smaller proportions [22,23,24,25].

Under healthy conditions, the gut microbiome contributes to immune homeostasis by modulating the gut microenvironment and coordinating immune responses [26,27]. Additionally, it performs essential metabolic functions, including the fermentation of polysaccharides, BA transformation, choline metabolism, energy extraction, and pathogen defense [28]. The GM is part of a dynamic ecosystem characterized by complex interactions involving metabolite exchange and immune signaling [15]. However, the microbiome is sensitive to various external and internal factors, including aging, dietary shifts, antibiotic use, environmental stress, immune status, and inflammation, which can disrupt its structure and diversity, leading to dysbiosis [22,23]. This imbalance may manifest as an excessive growth of certain bacterial species, the loss of beneficial microbes, or shifts in overall bacterial populations. Dysbiosis has been implicated in numerous metabolic disorders, including T2DM, obesity, metabolic syndrome, and MASLD [29,30].

### 2.2. Gut Microbiota in MASLD

Multiple studies in both human cohorts and animal models have demonstrated specific gut microbial alterations associated with the onset and progression of MASLD [31,32,33,34]. MASLD is often associated with reduced microbial diversity and a higher proportion of Gram-negative bacteria at the expense of Gram-positive bacteria [35]. These compositional shifts contribute to a pro-inflammatory intestinal environment and compromised gut barrier function, leading to increased microbial translocation and the exposure of the liver to bacterial endotoxins and other harmful metabolites [36,37].

Certain bacterial clades have been correlated with MASLD, including an increased abundance of *Pseudomonadota*, *Enterobacteriaceae*, and *Bacteroidaceae*, and the genera *Escherichia*, *Streptococcus*, *Fusobacterium*, *Bilophila*, *Bacteroides*, *Dorea*, and *Peptoniphilus*, along with a reduction in *Bacillota*, *Rikenellaceae*, and *Ruminococcaceae*, as well as the genera *Faecalibacterium*, *Coprococcus*, *Ruminococcus*, *Anaerosporobacter*, *Akkermansia*, *Blautia*, and *Eubacterium* [15,35]. Although *Pseudomonadota* typically comprise less than 5% of the gut microbiota, their expansion is considered a hallmark of dysbiosis and a key driver of metabolic and inflammatory responses [38]. An increased abundance of *Pseudomonadota* has also been associated with hepatic fibrosis in individuals with a normal body mass index [39]. Similarly, in patients with liver fibrosis, an increased abundance of *Ruminococcus* [40] and *Bacteroides vulgatus* [41] has been reported while obese patients with MASLD exhibit a relative increase in *Streptococcus* [42].

A recent study using ML algorithms identified sex-specific microbial signatures associated with MASLD, with *Christensenella* and *Limosilactobacillus* being the key taxa associated with MASLD in women and *Beduinibacterium* and *Anaerotruncus* in men [43]. Furthermore, advanced fibrosis in MASLD has been linked to a specific GM signature characterized by a high abundance of pathogenic bacteria, including *Fusobacterium* and *Escherichia-Shigella*, and the depletion of SCFAs-producing genera such as *Lachnospira* [44]. These microbial profiles, known as enterotypes, classify GM into distinct clusters based on the relative abundance of specific bacteria. Among these, the Prevotella/Bacteroides ratio has been highlighted as a predictive factor in dietary response, with individuals displaying higher ratios tending to achieve greater weight loss following lifestyle interventions [45]. Broadly, an excessive presence of enteric, inflammatory, and pathogenic bacterial genera, including *Escherichia*, *Streptococcus*, *Shigella*, *Dorea*, *Fusobacterium*, and *Enterococcus*, is closely linked to the onset and fibrotic progression of MASLD. Oppositely, less abundant bacterial genera in MASLD patients, such as *Lactobacillus*, *Akkermansia*, *Alistipes*, and *Eubacterium*, help regulate these same pathways, mitigating the disease progression and reducing the risk of its onset [15].

### 2.3. Conflicting Evidence in Gut Microbiota and MASLD Research

Despite these associations, identifying a robust and consistent microbial signature for MASLD remains challenging. Contradictory findings, such as the dual associations of *Blautia* and *Roseburia* with both disease progression and protection, highlight the complexity and heterogeneity of microbiome research [37]. For instance, while *Ruminococcus* has been positively linked to fibrosis in some studies [40], others report the opposite [41,46]. Similar inconsistencies have been reported for *Prevotella* spp. [40,47,48]. A recent study identified five bacterial genera that showed consistent differences in relative abundance across MASLD cohorts, but only *Lachnospiraceae* reached statistical significance, underscoring the absence of a universally reliable taxonomic marker [49].

Several factors likely contribute to these inconsistencies across investigations. First, the variability in study populations—due to heterogeneity in demographic (e.g., age, sex, and ethnicity), clinical (e.g., obesity, T2DM, and antibiotics use), and lifestyle characteristics—profoundly influences GM composition and may confound associations with MASLD [50]. Second, methodological differences in microbiota analysis—such as variations in DNA extraction protocols, targeted 16S rRNA gene regions (e.g., V3–V4 vs. V4–V5), sequencing platforms, and bioinformatics pipelines—can introduce substantial variability in taxonomic classification and relative abundance estimates [51]. Third, variations in MASLD definitions and diagnostic criteria—whether based on imaging techniques, liver biopsy, or non-invasive biomarkers—introduce heterogeneity in case identification and disease staging, thereby complicating comparisons across studies. Fourth, the predominance of cross-sectional designs in microbiota–MASLD research limits the ability to draw causal inferences. Moreover, temporal fluctuations in gut microbiota composition further complicate the interpretation of single-time-point analyses and may contribute to conflicting results [52].

The inherent complexity of the GM ecosystem and the phenomenon of functional redundancy indicate that taxonomic shifts may not directly reflect functional changes. Different microbial taxa can perform similar metabolic roles, meaning that taxonomic classification alone may be insufficient to explain disease mechanisms. This is particularly relevant for key microbial functions, such as butyrate and SCFA metabolic pathways, which are not exclusive to specific taxa. Therefore, relying solely on microbial diversity indices or compositional comparisons may fall short in elucidating the molecular mechanisms involved. To move beyond correlative findings, future studies must adopt multi-omics approaches and causal inference models to better understand the functional relevance of gut microbiota alterations in MASLD [53].

## 3. Microbiota-Derived Metabolites in MASLD Pathogenesis

Growing evidence highlights the critical role of the gut–liver axis in mediating communication between the gastrointestinal tract and the liver [54,55]. Disruptions in this axis, including microbiota dysbiosis and increased intestinal permeability, facilitate the translocation of microbial metabolites and components into the portal circulation, ultimately reaching the liver [56]. These microbial components, known as pathogen-associated molecular patterns (PAMPs), including lipopolysaccharide (LPS) and peptidoglycan, can activate pattern recognition receptors (PRRs), such as Toll-like receptors (TLRs), in Kupffer cells and hepatic stellate cells (HSCs), triggering inflammatory responses that contribute to liver injury and fibrogenesis [57,58].

Changes in the GM composition can lead to an alteration of the microbes-derived molecules and metabolites that influx in the systemic circulation. Several of these metabolites have been implicated in the pathogenesis of MASLD, including SCFAs, LPS, BA, choline, TMAO, and ammonia [13]. Through the production of these molecules, GM influences glucose and lipid metabolism, contributing to hepatic steatosis and metabolic dysfunction [32,53].

SCFAs, primarily acetate, propionate, and butyrate, support intestinal barrier function and exert anti-inflammatory effects. In contrast, LPS impairs gut integrity, promotes endotoxemia, and exacerbates hepatic inflammation. Altered BA profiles can disrupt enterohepatic circulation and metabolic regulation, while choline deficiency impairs the hepatic secretion of very-low-density lipoproteins (VLDLs), leading to triglyceride accumulation in the liver and contributing to MASLD development. Ammonia is increasingly recognized for its potential contribution to MASLD pathophysiology [13].

### 3.1. Gut Microbiota and SCFA Metabolism

Various GM members ferment complex carbohydrates, particularly those found in dietary fiber, into SCFAs, primarily acetate, propionate, and butyrate. While most SCFAs are utilized locally in the gut, a portion is absorbed by intestinal epithelial cells and enters the portal circulation, reaching the liver. There, they contribute to key metabolic processes: propionate serves as a substrate for gluconeogenesis, whereas acetate and butyrate are precursors for lipogenesis [59]. Among SCFAs, butyrate is particularly important as a fuel for enterocytes and for maintaining intestinal barrier integrity by increasing the expression of tight junction proteins and supporting the growth of *Lactobacillus* [60,61]. Improved gut barrier function reduces endotoxin leakage into circulation, thereby decreasing inflammatory cytokine activity in the liver and mitigating hepatic fat accumulation. SCFAs also exert systemic effects via the activation of G-protein-coupled receptors (GPCRs) expressed in enteroendocrine cells, adipocytes, and immune cells. GPCR activation triggers the release of peptide YY (PYY) and glucagon-like peptide-1 (GLP-1), which delay gastric emptying, enhance satiety, and promote hepatic lipid oxidation, thereby reducing steatosis [10,62,63,64].

Experimental models support the protective role of butyrate against liver steatosis. In mice, an increase in butyrate-producing bacteria prevented diet-induced hepatic fat accumulation [65]. In a clinical setting, lower fecal butyrate levels have been observed in diabetic patients [66]. Moreover, patients with T2DM following a high-fiber diet showed improved HbA1c levels, likely due to increased populations of acetate- and butyrate-producing bacteria and enhanced GLP-1 secretion. SCFA levels also vary with liver disease severity; patients with advanced fibrosis show elevated fecal acetate levels, while those with mild or moderate MASLD present higher butyrate and propionate concentrations [67]. However, in cirrhosis patients, circulating butyrate levels inversely correlate with inflammatory markers and endotoxin quantity [68]. These inconsistencies may reflect interindividual variability (age, diet, environmental factors) or technical challenges related to the volatile nature of SCFAs [13]. Despite this, SCFA supplementation in MASLD murine models has demonstrated beneficial effects, including reduced liver and adipose tissue inflammation, as well as shifts in the GM composition that favor SCFA-producing bacteria while reducing endotoxin-producing strains [69]. These findings suggest that SCFA supplementation could offer metabolic and hepatoprotective benefits.

Several commensal bacterial species enhance these effects. *Faecalibacterium Prausnitzii* produces significant SCFAs that provide energy for enterocytes and exert anti-inflammatory effects [70]. *F. prausnitzii* and its metabolites have shown protective effects against colitis in mice, increasing bacterial diversity and SCFA-producing bacteria while decreasing inflammatory markers [71]. Similarly, *Akkermansia muciniphila*, which colonizes the mucus layer of the intestine, interacts with the epithelial barrier via extracellular proteins and SCFA production (mainly acetate and propionate). In fact, this microorganism can counteract weight gain and immuno-metabolic disturbances by strengthening the gut barrier in animal models [72]. The abundance of these beneficial bacteria, particularly *Akkermansia*, can be increased through dietary polyphenols and flavonoids [73]. In mice, polyphenols promoted *A. muciniphila* growth, reducing inflammation, obesity, and insulin resistance [74,75]. Likewise, epigallocatechin-3-gallate (EGCG), a major green tea polyphenol, increased *A. muciniphila* abundance and butyrate levels while alleviating colonic inflammation in models of inflammatory bowel disease [76]. Flavonoids, plant-derived compounds, also promote SCFA production and modulate gut microbiota composition. In vitro studies have shown that certain flavonoids (baicalin, quercetin, icraiin, luteolin, amygdalin, and naringin) increase total volatile fatty acid production and potentially act as prebiotics, enhancing bacterial diversity [77]. Altogether, these findings support the therapeutic potential of flavonoid-rich and polyphenol-rich diets in modulating gut microbiota, increasing beneficial SCFA-producing bacteria, and ameliorating metabolic and inflammatory diseases such as MASLD.

### 3.2. Gut Microbiota and BA Metabolism

The bidirectional relationship between the GM and BAs is well established. On one hand, BAs regulate microbial populations by preventing bacterial overgrowth and preserving intestinal barrier integrity; on the other hand, the GM modulates the BA composition through enzymatic transformation [78]. Since BAs have a major impact on the host metabolism and immune function through farnesoid X receptor (FXR) and membrane-associated GPCR5 signaling, gut dysbiosis or BA imbalance may contribute to the development of metabolic diseases [79,80]. Within the gut, primary BAs are deconjugated and dehydroxylated by the GM into more hydrophobic secondary BAs. These are reabsorbed in the distal ileum and returned to the liver via enterohepatic circulation [81,82]. Secondary BAs can activate FXR and GPCR5, thereby modulating metabolic and immune pathways, including the inflammatory response [83,84].

Alterations in BA profiles have been associated with liver disease progression. Notably, specific BA compositions have been linked to fibrosis in MASLD via the activation of the NLRP3 inflammasome [85]. Elevated ratios of circulating conjugated chenodeoxycholic acid to muricholic acid correlated with increased histological severity and fibrosis in MASH patients [86]. Additionally, the levels of 7-hydroxy-4-cholesten-3-one (C4), a marker for de novo BA synthesis, were increased in the serum of MASH patients and were also associated with alterations in the GM [87].

The GM can regulate BA metabolism through microbial enzymes such as bile salt hydrolase (BSH) and hydroxysteroid dehydrogenases (HSDHs), which facilitate the deconjugation and oxidation/epimerization of primary BAs [88]. Microbial species that overexpress BSH, such as *Lactobacillus casei*, help reduce hepatic steatosis, cholesterol accumulation, and lipid metabolism dysfunction in in vitro models [89]. Similarly, *Eubacterium*, *Ruminococcus*, and *Bacteroides*, which express HSDHs [90], could potentially promote ursodeoxycholic acid (UDCA) formation [91]. UDCA has demonstrated hepatoprotective effects by activating AMPK, reducing oxidative stress, and attenuating hepatic inflammation [92]. Moreover, beneficial bacteria such as *Akkermansia muciniphila* and *Bifidobacterium bifidum* help prevent the development of MASLD in mouse models by regulating FXR signaling, leading to reduced weight gain, improved insulin sensitivity, and decreased liver fat accumulation [93].

### 3.3. Gut Microbiota and Choline Metabolism

Choline, an essential component of the cell membrane, is mainly obtained from dietary sources (e.g., red meat, eggs, cheese, peanuts), although it can be synthesized by humans de novo, but at insufficient levels. In the liver, choline is involved in the production of VLDL, preventing the hepatic accumulation of triglycerides. For this reason, choline-deficient diets have been used in animal models to induce MASH [94].

Choline can be converted to trimethylamine (TMA) by GM, which is absorbed by intestinal epithelial cells. In the liver, TMA is oxidized to generate TMAO [95]. Microbial metabolites like TMAO, which are elevated in individuals with steatotic liver disease, contribute to lipid deposition in hepatocytes by weakening the intestinal barrier and activating the TLR4/NF-κB pathway [96]. TMAO has been linked to impaired glucose tolerance and the progression of MASLD [97], with elevated serum TMAO levels correlating with both hepatic steatosis severity in MASLD patients [98] and MASH in T2DM patients [99]. In addition, metabolomics studies in humans identified TMAO as a predictor of thrombotic events, linked to its contribution to platelet hyperreactivity [100].

### 3.4. Gut Microbiota and Ammonia Metabolism

Urea synthesis—the main process for ammonia detoxification—takes place in the liver. This cycle is impaired in MASLD patients due to a decrease in the activity and expression of urea cycle enzymes that leads to reduced ammonia degradation and thus hyperammonemia. Ammonia can directly activate HSCs, potentially promoting liver fibrosis [101]. It is also produced by gut bacteria from amino acids, meaning that the GM composition influences circulating ammonia [102]. Hyperammonemia is considered a potential indicator of liver disease severity. In MASH preclinical models, dietary intervention resulted in the restoration of the normal urea cycle enzyme activity and liver fat reduction [103].

In addition to its role in the liver, ammonia is a neurotoxic compound that crosses the blood–brain barrier and plays a major role in hepatic encephalopathy, though it is not the only factor [104].

## 4. Influence of Dietary Patterns in MASLD

Diet is one of the most powerful and modifiable modulators of GM composition and function. Accumulating evidence highlights the crucial role of dietary patterns not only in shaping microbial diversity and metabolic activity but also in influencing the development and progression of MASLD [105,106]. The Western diet, typically high in saturated fats, refined carbohydrates, and low in fiber, is associated with gut dysbiosis, reduced microbial diversity, and the overgrowth of pro-inflammatory taxa [15]. Moreover, processed foods, which often contain additives, preservatives, and dietary emulsifiers, disrupt the balance of gut microbiota, increasing intestinal permeability, exacerbating metabolic dysfunctions, and promoting fat accumulation in the liver [36,107]. In contrast, dietary patterns rich in fiber, polyphenols, and unsaturated fats, such as the Mediterranean diet, promote the growth of beneficial SCFA-producing bacteria, support intestinal barrier integrity, and exert anti-inflammatory effects [108]. These diet-induced microbial changes influence key host metabolic pathways, including lipid and glucose homeostasis, bile acid metabolism, and immune regulation, all of which are implicated in MASLD pathogenesis [109,110]. Moreover, specific microbial metabolites, such as SCFAs, TMAO, and BAs, act as mediators between dietary inputs and hepatic outcomes. Understanding the diet–microbiota–liver axis is therefore essential for identifying preventive and therapeutic strategies against MASLD.

### 4.1. Western Diet and Dysbiosis

The Western diet, characterized by a high intake of saturated fats, refined sugars, red and processed meats, and ultra-processed foods, alongside a low consumption of dietary fiber, is strongly associated with gut microbiota dysbiosis and the development of MASLD. This dietary pattern promotes a shift in microbial composition, favoring the growth of pro-inflammatory and endotoxin-producing bacteria such as *Escherichia coli*, *Streptococcus*, and *Dorea*, while reducing beneficial taxa like *Faecalibacterium prausnitzii*, *Akkermansia muciniphila*, and *Ruminococcus* [108]. Dysbiosis induced by a Western diet leads to a reduced production of SCFAs, particularly butyrate, which impairs intestinal barrier integrity and increases gut permeability [111,112]. This facilitates the translocation of PAMPs, such as LPS, into the portal circulation, where they trigger hepatic inflammation through TLR activation on Kupffer cells and HSCs. Inflammatory signaling and immune activation driven by this gut–liver crosstalk contribute to hepatic steatosis, ballooning, and fibrotic progression in MASLD. Moreover, Western dietary patterns alter bile acid metabolism by promoting the abundance of bacterial species with BSH and HSDH activity, which modify the composition of primary and secondary bile acids [108]. These changes can disrupt FXR and TGR5 signaling, contributing to insulin resistance, lipid dysregulation, and hepatic inflammation (see Figure 1).

A hallmark of the Western diet is excessive fructose intake, mainly through sugar-sweetened beverages and processed foods. High fructose consumption is a well-recognized driver of hepatic steatosis, de novo lipogenesis, and gut microbiota disruption [113]. Fructose is primarily absorbed in the small intestine, but when consumed in large amounts, unabsorbed fructose reaches the colon, where it is rapidly fermented by colonic bacteria. This process promotes the overgrowth of fermentative and ethanol-producing bacteria, such as certain *Clostridium* and *Escherichia* species. Importantly, fructose-induced dysbiosis contributes to increased intestinal permeability and the translocation of endotoxins like LPS, exacerbating hepatic inflammation through TLR4 signaling. Additionally, fructose enhances the luminal production of endogenous ethanol and reactive oxygen species, further contributing to mitochondrial dysfunction and oxidative stress in hepatocytes [15,36,109,110].

Animal studies have consistently shown that high-fat, high-sugar diets induce dysbiosis and rapidly lead to steatosis and inflammation [114,115,116]. In murine models, FMT from Western diet-fed donors is sufficient to induce hepatic steatosis in germ-free mice, even in the absence of direct dietary exposure [117]. This highlights the causal role of gut microbial alterations in MASLD pathogenesis and the deleterious synergy between poor dietary quality and microbiota composition.

### 4.2. Mediterranean Diet and Microbiota Restoration

In contrast to the Western diet, the Mediterranean diet is associated with a protective gut microbial profile and improved liver health outcomes. It emphasizes the intake of unprocessed, whole-plant foods, olive oil (particularly extra-virgin olive oil, EVOO), and dairy products, with a moderate consumption of poultry and fish, while limiting red and processed meats [108]. This dietary pattern is naturally rich in dietary fiber, polyphenols, monounsaturated fats, and *n*-3 polyunsaturated fatty acids, all of which contribute to its anti-inflammatory and antioxidant effects.

The Mediterranean diet has been shown to favorably modulate the gut microbiota. Two interventional studies demonstrated that adherence to this diet increases the abundance of beneficial bacteria such as *Faecalibacterium prausnitzii* and *Roseburia* spp., while reducing the presence of potentially pathogenic or pro-inflammatory species like *Ruminococcus gnavus*, *Collinsella aerofaciens*, and *Ruminococcus torques* [118,119]. These taxonomic shifts are associated with an enhanced production of SCFAs and reduced levels of harmful microbial metabolites such as ethanol, para-cresols, and carbon dioxide [118]. In addition, EVOO has been specifically linked to improved postprandial glycemic control, partly by reducing low-grade endotoxemia linked to increased gut permeability [120]. The anti-inflammatory properties of EVOO, along with its polyphenol content, likely contribute to this effect, supporting intestinal integrity and modulating microbial composition toward eubiosis.

Polyphenols, abundant in olive oil, red wine, berries, and leafy greens, also play a crucial role in modulating gut microbiota. They act as prebiotics, selectively promoting the proliferation of *Akkermansia muciniphila*, a mucin-degrading bacterium associated with improved metabolic profiles, reduced fat mass, and enhanced intestinal barrier function [73]. Polyphenol-rich diets have been shown to increase microbial diversity and lower circulating levels of pro-inflammatory markers in both animal and human studies [74,75,76,121,122].

A metagenomic shotgun sequencing study of long-term microbiome data from 307 individuals found that the Mediterranean diet modulated 36 distinct microbial metabolic pathways, enhancing microbial functions related to SCFA production and fiber degradation [123]. Interestingly, the protective association between an adherence to the Mediterranean diet and cardiometabolic disease risk was particularly pronounced in individuals with a low abundance of *Prevotella copri*, a species often linked to pro-inflammatory responses and metabolic dysfunction.

### 4.3. High-Fiber Diets and SCFA Production

Dietary fiber plays a central role in shaping gut microbiota composition and function, and its intake is inversely associated with the risk and severity of metabolic disorders, including MASLD. Unlike digestible carbohydrates, dietary fibers reach the colon intact, where they are fermented by specific microbial populations into SCFAs [32,53]. These microbial metabolites exert local and systemic effects that support intestinal and hepatic health.

SCFA production is closely tied to the abundance of specific bacterial taxa. Genera such as *Faecalibacterium*, *Roseburia*, *Blautia*, *Coprococcus*, and *Ruminococcus* are well-known fiber degraders and efficient butyrate producers [124,125,126]. Diets rich in fermentable fibers enhance the growth of these bacteria and are associated with improved gut barrier function, reduced intestinal permeability, and a decrease in endotoxin translocation to the liver [108].

In MASLD, the reduced intake of dietary fiber is associated with diminished SCFA production and dysbiosis, leading to increased intestinal permeability, low-grade inflammation, and hepatic lipid accumulation. Animal models have shown that supplementation with fermentable fibers or the direct administration of SCFAs ameliorates steatosis and reduces markers of hepatic inflammation [69]. Furthermore, clinical studies have demonstrated that high-fiber diets in patients with metabolic syndrome or T2DM improve glycemic control and modulate the gut microbiota toward an SCFA-producing profile, likely contributing to the amelioration of hepatic fat content [127,128,129].

### 4.4. Polyphenols, Flavonoids, and Gut–Liver Interactions

Polyphenols, a diverse class of plant-derived bioactive compounds, have emerged as important dietary modulators of gut microbiota composition and metabolic health. Widely present in fruits, vegetables, tea, coffee, cocoa, and olive oil, polyphenols—including flavonoids, phenolic acids, stilbenes, and lignans—exert antioxidant, anti-inflammatory, and immunomodulatory effects. Their low bioavailability in the upper gastrointestinal tract allows them to reach the colon, where they interact directly with the gut microbiota, influencing both microbial composition and function.

Polyphenols serve as substrates for microbial metabolism and act as selective growth promoters of beneficial bacteria, particularly *Akkermansia muciniphila* and *Faecalibacterium prausnitzii*, both of which are associated with an improved gut barrier function, SCFA production, and anti-inflammatory activity [70,130]. An increased abundance of *A. muciniphila* has been linked to improved insulin sensitivity, reduced adiposity, and the attenuation of hepatic steatosis in animal models [72]. Several preclinical and human studies have demonstrated that polyphenol-rich diets increase gut microbial diversity and promote the production of SCFAs, while simultaneously reducing the abundance of endotoxin-producing taxa. For example, cranberry and grape-derived polyphenols have been shown to stimulate *A. muciniphila* proliferation and reduce metabolic inflammation and obesity-related parameters in mice [74,75]. Similarly, supplementation with EGCG—the primary catechin in green tea—ameliorated colonic inflammation in the mouse models of IBD by enhancing butyrate production and increasing *A. muciniphila* abundance [76].

Flavonoids, a prominent subclass of polyphenols, have also been shown to modulate gut microbiota and support intestinal homeostasis. In vitro studies have demonstrated that flavonoids such as baicalin, quercetin, icariin, luteolin, amygdalin, and naringin can enhance the production of total volatile fatty acids (VFAs), acting as prebiotic-like compounds that improve bacterial diversity and stimulate SCFA synthesis [77].

Through these mechanisms, polyphenols and flavonoids not only contribute to the restoration of gut microbial balance but also modulate key host pathways involved in MASLD pathogenesis, including inflammation, oxidative stress, lipid metabolism, and insulin resistance.

### 4.5. Other Dietary Components and Microbial Modulation

Beyond fiber and polyphenols, other dietary components, such as sweeteners and food additives, also play a significant role in shaping gut microbiota composition and influencing liver health [131].

Artificial sweeteners such as sucralose, saccharin, and aspartame, frequently used in “sugar-free” processed foods, have been shown to alter gut microbial communities and impair glucose metabolism [107]. These compounds may reduce the abundance of SCFA-producing bacteria and promote the growth of strains linked to metabolic endotoxemia.

Emulsifiers (e.g., carboxymethylcellulose and polysorbate-80), preservatives, and other synthetic food additives have been reported to compromise gut barrier function and disturb microbial composition [132]. In murine models, chronic exposure to emulsifiers led to decreased *Akkermansia muciniphila* abundance, increased mucosal inflammation, and the development of metabolic syndrome features [133]. Although human data remain limited, these findings raise concerns about the long-term effects of ultra-processed food consumption on gut–liver homeostasis.

### 4.6. Diet–Microbiota Interactions in MASLD Progression

A growing body of evidence supports a causal relationship between dietary patterns, gut microbiota composition, and the development and progression of MASLD.

Observational studies in diverse populations have shown that dietary quality is strongly associated with gut microbiota profiles and MASLD risk. For example, individuals with high adherence to the Mediterranean diet exhibit increased microbial diversity, enrichment of SCFA-producing genera (*Faecalibacterium*, *Roseburia*), and lower hepatic steatosis scores compared to those following Western dietary patterns [108,134]. Clinical trials have confirmed that dietary interventions rich in fiber and polyphenols not only shift microbiota composition but also reduce hepatic fat content and improve metabolic markers. In patients with obesity and MASLD, high-fiber or polyphenol-enriched diets have led to increases in *Akkermansia muciniphila* and *Bifidobacterium*, coupled with improvements in insulin sensitivity and liver function tests [135].

Rodent models have been instrumental in demonstrating mechanistic links between diet, microbiota, and MASLD. Mice fed high-fat, high-fructose, or choline-deficient diets develop microbiota alterations similar to those observed in human MASLD, including increased *Pseudomonadota* and reduced butyrate-producing taxa [115,136,137]. These models reproduce the key histological features of MASLD, such as steatosis, lobular inflammation, and fibrosis, while allowing for the controlled manipulation of dietary components and microbial exposures [138]. Importantly, dietary reversal in these models, such as transitioning from a Western to a Mediterranean-style or high-fiber diet, leads to the restoration of microbial diversity and the attenuation of liver damage, supporting a direct role of diet–microbiota interactions in disease modulation [139].

FMT studies provide strong evidence for the transmissibility of MASLD-associated phenotypes via the gut microbiota. In one notable experiment, germ-free mice colonized with fecal material from MASLD patients developed hepatic steatosis and metabolic dysfunction despite being maintained on a standard diet, whereas mice receiving feces from healthy donors were protected. These findings confirm that the altered microbial communities seen in MASLD are not merely a consequence of disease but can actively drive its development [140].

Despite compelling evidence, several limitations hinder the translation of microbiome–diet research into clinical practice. First, most studies rely on 16S rRNA sequencing, which provides limited taxonomic resolution and no direct insight into microbial function [141]. While metagenomic and metabolomic approaches offer a deeper view, they are costly and not yet widely standardized. Second, dietary intake is often assessed through self-reported tools that are prone to recall bias [142]. Finally, interindividual factors—including sex, age, genetics, medication use, and lifestyle—can confound associations and obscure causal inferences. As a result, future research should prioritize well-controlled, longitudinal studies using multi-omics integration and personalized dietary interventions. Incorporating AI-based tools may help overcome heterogeneity and optimize individualized dietary strategies based on microbiome profiles.

## 5. Artificial Intelligence in Lifestyle Interventions for MASLD

Lifestyle modification remains the cornerstone of MASLD prevention and management, yet sustained behavioral change is difficult to achieve in routine clinical practice. In this context, AI is emerging as a transformative tool to enhance the effectiveness, personalization, and scalability of lifestyle interventions [17]. By leveraging large-scale, multidimensional datasets (including clinical, dietary, microbiome, and metabolic profiles), AI models can identify complex patterns and generate predictive insights that go beyond traditional statistical methods [143,144,145,146].

ML algorithms have shown promise in predicting MASLD risk, stratifying patients by disease severity, and identifying microbial or metabolic signatures associated with disease progression [41,147,148]. More recently, AI-driven platforms have been applied to support personalized nutrition and behavioral change, offering dynamic dietary recommendations, monitoring adherence in real time, and predicting individual responses to interventions based on baseline characteristics [149]. These developments are particularly relevant in MASLD, where the variability in treatment response is high, and where the gut microbiota serves as both a biomarker and therapeutic target. Integrating AI into lifestyle intervention strategies may help bridge the gap between precision medicine and population-level implementation, offering new opportunities for the early detection, targeted prevention, and tailored management of MASLD.

### 5.1. AI-Driven Dietary and Exercise Programs

Recent scientific studies have evaluated AI-based programs aimed at promoting lifestyle modifications and preventing chronic diseases. Specifically, some studies show the role of AI in disease prediction, treatment improvement, and real-time monitoring based on nutritional intake data [150,151]. In the context of MASLD, where sustained lifestyle change is essential but often difficult to achieve, AI-driven systems offer scalable, personalized, and adaptive solutions that go beyond traditional approaches. Intelligent digital platforms can provide real-time dietary feedback and generate individualized meal plans tailored to health goals such as weight loss, glycemic control, or improved lipid profiles [152,153,154,155].

AI is being investigated in MASLD to develop reproducible, quantitative, and automated methods to enhance patient stratification and discover new biomarkers [18]. Different applications of AI and machine learning algorithms in MASLD, particularly in analyzing electronic health records, digital pathology, and imaging data, are being developed to support diagnosis, risk assessment, and treatment planning [156,157]. Furthermore, microbiota-based ML approaches are emerging as promising tools for disease classification and personalized therapy selection [147]. A key strength of AI-driven nutrition programs lies in their capacity to integrate multi-omics data—such as metagenomics, metabolomics, and transcriptomics—using models like deep learning and random forests. This enables the modeling of dynamic interactions between diet, gut microbiota, and host metabolism. AI systems can process large volumes of individualized health data to predict responses to specific nutrients and dietary patterns. For instance, predictive algorithms can estimate the impact of high-fiber diets on the abundance of beneficial microbes such as *Akkermansia muciniphila* and *Faecalibacterium prausnitzii*. These models can also account for genetic predisposition, metabolic flexibility, and behavioral adherence, allowing for highly tailored dietary interventions (see Figure 2).

Despite their promise, few AI-based lifestyle programs have been specifically validated in patients with diagnosed MASLD. Nevertheless, due to the pathophysiological overlap between MASLD and other metabolic conditions—such as obesity, T2DM, and metabolic syndrome—the beneficial effects observed in these populations are likely to be applicable. Future research should aim to include liver-specific outcomes, such as hepatic steatosis and fibrosis scores, to better assess the direct impact of AI-driven interventions on MASLD progression.

### 5.2. Conversational Agents

Conversational agents—commonly referred to as chatbots—are AI-driven systems designed to simulate human dialog and provide interactive, real-time communication with users through natural language processing (NLP). In the context of health promotion, chatbots have shown significant potential in supporting behavior change by offering personalized guidance, continuous motivation, and education without the need for constant human intervention. AI-powered chatbots have proven effective in encouraging healthy behaviors, such as physical activity, fruit and vegetable consumption, smoking cessation, medication adherence, and improved sleep duration and quality [16,17]. While AI chatbots show potential in promoting healthy eating habits, current evidence is insufficient to confirm their effectiveness in weight loss [158]. Well-designed interventions are necessary to determine whether chatbots can genuinely support long-term weight loss.

These chatbots, considered conversational agents or virtual assistants, employ behavior change strategies such as goal setting, progress monitoring, and real-time feedback, delivering personalized interventions through accessible platforms and devices. In addition, emerging research collectively shows that AI is a greater option for transforming the methods and tools used for dietary evaluation, thereby reducing manual efforts and introducing more accurate and efficient approaches [159]. In this sense, one study evaluates the feasibility and efficacy of a 12-week lifestyle intervention led by an AI chatbot that generates a personalized guide on physical activity and a Mediterranean-style diet for each participant. Results showed significant improvements in participants’ physical activity, higher Mediterranean diet scores, and a decrease in waist circumference [160]. Regarding dietary adherence to the Mediterranean diet, a recent study suggests that an automated smartphone application can effectively monitor and evaluate users’ adherence based on an imaging system with the same accuracy as a specialist [161].

Chatbots offer an opportunity to democratize access to health interventions due to their cost-effectiveness, accessibility, and scalability, while also complementing healthcare professionals in preventive programs. However, while AI and ML offer promising advancements in precision nutrition, researchers also caution about the potential risks associated with their use, and the need to develop a holistic risk management framework for AI/ML systems is crucial to ensure their safe and effective deployment in high-stakes medical settings [162].

### 5.3. Future Perspectives

The future of personalized dietary management for MASLD is closely linked to advances in AI, multi-omics technologies, and the integration of clinical and behavioral data into predictive models. These innovations are paving the way for highly individualized nutritional strategies that optimize hepatic and metabolic outcomes while considering patient-specific variables. AI-driven machine learning models, including deep neural networks and random forest algorithms, are increasingly capable of processing large-scale datasets from metagenomics, metabolomics, and transcriptomics to generate highly personalized dietary recommendations. The development of digital platforms integrating these models could allow the real-time monitoring of microbiota and metabolic changes in response to dietary interventions.

Omics technologies are also essential for identifying specific biomarkers associated with MASLD and predicting responses to different dietary interventions. Their integration into clinical practice could facilitate patient stratification, enabling more targeted interventions based on individual metabolic profiles.

Despite the great potential of AI in personalized nutrition, several challenges must be overcome before these technologies can be translated into routine clinical care. These include the standardization and interoperability of omics data, the need for longitudinal studies to assess the durability and safety of AI-guided interventions, and the inclusion of contextual variables such as physical activity, medication use, alcohol consumption, and sleep quality in predictive models. Ethical and regulatory frameworks must also be established to ensure data security, algorithm transparency, and equitable access to these tools.

Ultimately, the integration of AI with multi-omics approaches holds immense potential to transform the management of MASLD through the development of personalized, adaptive, and mechanistically informed dietary interventions. These technologies will not only enhance our ability to identify novel biomarkers but also offer new therapeutic avenues that move beyond uniform, generalized strategies, ushering in a new era of precision hepatology.

## 6. Conclusions

The complex interplay between dietary habits, gut microbial composition, and liver health has gained recognition as a key element in MASLD pathogenesis and progression. Specific dietary patterns, particularly the Western diet, can disrupt gut microbial homeostasis, leading to intestinal permeability, systemic inflammation, and metabolic dysregulation, which contribute to hepatic steatosis and fibrosis. In contrast, diets rich in fiber, polyphenols, and unsaturated fats, such as the Mediterranean diet, have been shown to foster a more favorable gut microbiota profile, increasing the abundance of SCFA-producing bacteria, reducing endotoxemia, and improving liver-related outcomes.

Beyond dietary factors, AI-based tools, including machine learning models and conversational agents, are emerging as promising strategies to support personalized nutrition and behavior change. These systems can provide tailored dietary and exercise recommendations, predict individual responses to interventions, and enhance long-term adherence. Although clinical evidence in MASLD-specific populations is still limited, findings from related metabolic conditions suggest that these technologies could significantly improve adherence and clinical outcomes when properly implemented.

Looking ahead, the integration of AI with omics data (such as metagenomics, metabolomics, and transcriptomics) holds great promise for advancing precision nutrition strategies in MASLD. Such approaches can move the field beyond generalized recommendations toward personalized, mechanism-driven interventions tailored to each patient’s metabolic and microbial profile. Nevertheless, key challenges remain, including the need for standardized clinical endpoints, data harmonization, long-term validation, and ethical frameworks, to ensure the safe and equitable application of AI in healthcare. Addressing these gaps will be essential to fully realize the potential of dietary modulation and digital tools in the prevention and management of MASLD.

In conclusion, improving MASLD management will likely require a multifaceted approach that combines dietary modulation and digital health tools. Continued research and clinical validation will be essential to translate these innovations into practical, effective interventions that can be implemented in real-world settings.

## Figures and Tables

**Figure 1 nutrients-17-01580-f001:**
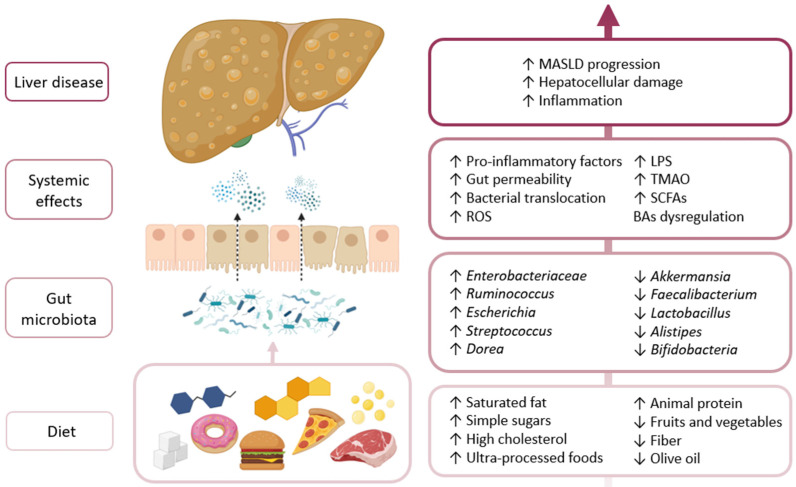
Diet directly influences microbial composition, leading to an increase in pro-inflammatory bacteria and a decrease in beneficial microbes. This imbalance is reflected systemically by enhanced intestinal permeability and elevated levels of inflammatory mediators, which contribute to the progression of liver disease.

**Figure 2 nutrients-17-01580-f002:**
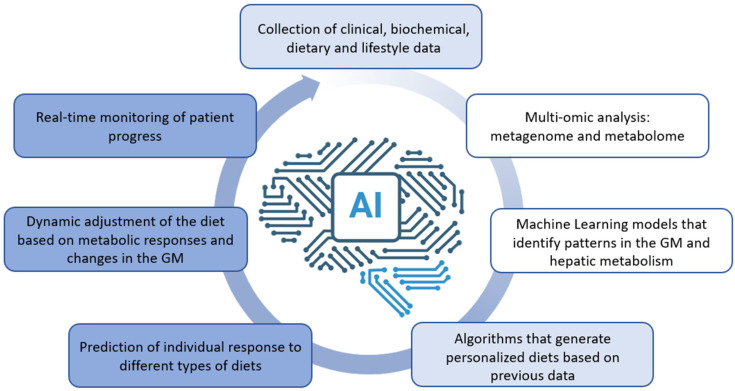
Data collection, management, and analysis are performed by experts, enabling the development of algorithms to personalize lifestyle interventions for the patient. These tools are accessible to the patient in real-time, facilitating dynamic adjustments based on their ongoing progress.

## Data Availability

Not applicable.

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
