# Peer review of "Unraveling MASLD: The Role of Gut Microbiota, Dietary Modulation, and AI-Driven Lifestyle Interventions"

_nutrients, 2025, doi:10.3390/nu17091580_

Round 1

Reviewer 1 Report

Comments and Suggestions for Authors

Dear Editor and Authors,

I read and assessed this quite interesting review work titled "Unraveling MASLD: The role of gut microbiota, dietary modulation, and AI-driven lifestyle interventions" in which the authors expore the role and interconnections between metabolic-associated steatotic liver disease (MALSD) and the gut microbiome and how these interactions can be modulated with the use of AI inspired lifestyle and dietary modifications.

Overall after reading this work I found it to be quite thorough and detailed presenting in a comprehensive, well organized manner all main points. The figures are detailed and explanatory.

As a Mediteranean person I was quite satisfied by the section on the effect of the Mediteranean diet on gut microbiota and liver restoration/health! 

Overall this is a well written, comprehensive and informative review that explores well all aspects of the pathology of MASLD and its association with the enteric microbiome. In addition the section on AI influenced/guided lifestyle modifications was interesting and hopefull that in time they will play a bigger role in patient management/self-aid.

Author Response

Comment 1: Dear Editor and Authors,

I read and assessed this quite interesting review work titled "Unraveling MASLD: The role of gut microbiota, dietary modulation, and AI-driven lifestyle interventions" in which the authors expore the role and interconnections between metabolic-associated steatotic liver disease (MALSD) and the gut microbiome and how these interactions can be modulated with the use of AI inspired lifestyle and dietary modifications.

Overall after reading this work I found it to be quite thorough and detailed presenting in a comprehensive, well organized manner all main points. The figures are detailed and explanatory.

As a Mediteranean person I was quite satisfied by the section on the effect of the Mediteranean diet on gut microbiota and liver restoration/health! 

Overall this is a well written, comprehensive and informative review that explores well all aspects of the pathology of MASLD and its association with the enteric microbiome. In addition the section on AI influenced/guided lifestyle modifications was interesting and hopefull that in time they will play a bigger role in patient management/self-aid.

Response 1: We thank the reviewer for their thoughtful and positive feedback. We are glad that the organization and detail of the review, including the figures and the discussion on the Mediterranean diet, were well received. We also appreciate the encouraging comments regarding the potential of AI-driven lifestyle interventions. Thank you again for your kind words and support.

Reviewer 2 Report

Comments and Suggestions for Authors

Authors reviewed about the role of gut microbiota, dietary modulation and AI0driven lifestyle intervention in MASLD. MASLD is a leading chronic liver disease, and its complex pathogenesis is a serious issue to be resolved. This review is comprehensive and well-written.  

Author Response

Comment 1: Authors reviewed about the role of gut microbiota, dietary modulation and AI0driven lifestyle intervention in MASLD. MASLD is a leading chronic liver disease, and its complex pathogenesis is a serious issue to be resolved. This review is comprehensive and well-written. 

Response 1: We sincerely thank the reviewer for their positive evaluation of our manuscript and their kind comments regarding the comprehensiveness and clarity of our review. We truly appreciate their encouraging feedback and are glad that the manuscript was found to be valuable and well-prepared.

Reviewer 3 Report

Comments and Suggestions for Authors

Thank you for submitting the manuscript "Unraveling MASLD: The role of gut microbiota, dietary modulation, and AI-driven lifestyle interventions" to Nutrients.

The manuscript is very informative, with an in-depth review of the relationship between MASLD, gut microbiota, dietary interventions, and AI-driven tools.

- The first issue I have with this review is that authors who propose to review a microbiota topic should use the correct phylum nomenclature, since for consistency reasons many names have been changed and the manuscript contains the old name.

- Some sections are too long and could be subdivided with more specific subtitles. For example, within the section on diets, the topics on fiber, polyphenols, and additives could be separated with more visual prominence.

- The review could benefit from a more critical analysis of conflicting studies, as is the case with inconsistencies in microorganisms presented but without presenting the reason for these discrepancies.

- It is also necessary to evaluate and clearly indicate the limited clinical evidence from the review

Author Response

We would like to thank the reviewer for the critical and constructive evaluation of our manuscript entitled "Unraveling MASLD: The Role of Gut Microbiota, Dietary Modulation, and AI-Driven Lifestyle Interventions". We appreciate the opportunity to revise and improve our work based on these insightful comments.

Comment 1: The first issue I have with this review is that authors who propose to review a microbiota topic should use the correct phylum nomenclature, since for consistency reasons many names have been changed and the manuscript contains the old name.

Response 1: We thank the reviewer for this important observation. We have thoroughly revised the manuscript to adopt the updated taxonomic nomenclature according to Oren A et al., 2021 (doi:10.1099/ijsem.0.005056). Specifically, "Firmicutes" has been replaced by "Bacillota", "Bacteroidetes" by "Bacteroidota", "Actinobacteria" by "Actinomycetota", "Proteobacteria" by "Pseudomonadota", "Fusobacteria" by "Fusobacteriota", and "Verrucomicrobia" by "Verrucomicrobiota" throughout the text. Additionally, an explanatory note has been added at first mention to enhance reader comprehension.

Comment 2: Some sections are too long and could be subdivided with more specific subtitles. For example, within the section on diets, the topics on fiber, polyphenols, and additives could be separated with more visual prominence.

Response 2: We thank the reviewer for this constructive suggestion. In response, we have revised the text to improve readability and structure, organizing the information more clearly to facilitate comprehension.

Comment 3: The review could benefit from a more critical analysis of conflicting studies, as is the case with inconsistencies in microorganisms presented but without presenting the reason for these discrepancies.

Response 3: We fully agree with the reviewer’s observation. We have added a new subsection "2.3 Conflicting evidence in gut microbiota and MASLD research", in which we discuss the potential causes of inconsistencies across studies. These include variability in study populations, differences in microbiota analysis techniques, heterogeneity in MASLD diagnostic criteria, cross-sectional design limitations, influences of external factors, and functional redundancy among microbial taxa.

Comment 4: It is also necessary to evaluate and clearly indicate the limited clinical evidence from the review.

Response 4: In response to this comment, we have analyzed limitations related to the quality of available studies, the short duration of interventional trials, the early development of AI-based interventions, and the need for personalized approaches. We have highlighted the need for robust, longitudinal, and mechanistically based clinical trials in the field of MASLD.